# Gut Microbial-Derived Metabolomics of Asthma

**DOI:** 10.3390/metabo10030097

**Published:** 2020-03-06

**Authors:** Kathleen A. Lee-Sarwar, Jessica Lasky-Su, Rachel S. Kelly, Augusto A. Litonjua, Scott T. Weiss

**Affiliations:** 1Channing Division of Network Medicine, Brigham and Women’s Hospital and Harvard Medical School, Boston, MA 02115, USA; rejas@channing.harvard.edu (J.L.-S.); rachel.kelly@channing.harvard.edu (R.S.K.); scott.weiss@channing.harvard.edu (S.T.W.); 2Division of Rheumatology, Immunology and Allergy, Brigham and Women’s Hospital and Harvard Medical School, Boston, MA 02115, USA; 3Division of Pediatric Pulmonary Medicine, Golisano Children’s Hospital at Strong, University of Rochester Medical Center, Rochester, NY 14612, USA; augusto_litonjua@urmc.rochester.edu

**Keywords:** microbiome, metabolomics, asthma, short chain fatty acids, polyunsaturated fatty acids, bile acids, tryptophan, sphingolipids

## Abstract

In this review, we discuss gut microbial-derived metabolites involved with the origins and pathophysiology of asthma, a chronic respiratory disease that is influenced by the microbiome. Although both gut and airway microbiomes may be important in asthma development, we focus here on the gut microbiome and metabolomic pathways involved in immune system ontogeny. Metabolite classes with existing evidence that microbial-derived products influence asthma risk include short chain fatty acids, polyunsaturated fatty acids and bile acids. While tryptophan metabolites and sphingolipids have known associations with asthma, additional research is needed to clarify the extent to which the microbiome contributes to the effects of these metabolites on asthma. These metabolite classes can influence immune function in one of two ways: (i) promoting growth or maturity of certain immune cell populations or (ii) influencing antigenic load by enhancing the number or species of specific bacteria. A more comprehensive understanding of how gut microbes and metabolites interact to modify asthma risk and morbidity will pave the way for targeted diagnostics and treatments.

## 1. Introduction: Microbiome-Metabolome Associations in Asthma

Asthma and other allergic diseases have well known associations with early life environmental exposures that modify the gut microbiota, such as living on a farm, mode of delivery, breastfeeding status and having a dog in the home [1]. As mounting animal and human data point to a prominent role of the gut microbiome in asthma development [2], relevant metabolomic mechanisms behind this association are beginning to be elucidated [3]. Integration of metabolomic data with gut microbiome data has been particularly fruitful in understanding the gut-lung axis as it pertains to asthma. Here, we review a set of metabolites and metabolite groups that appear to link the gut microbiota with asthma development and pathophysiology and immune system ontology. Some of these classes, such as short chain fatty acids, are relatively well-studied and understood, while others, including the sphingolipids, include more numerous metabolites with less straightforward relationships to asthma and allergy. While we focus here on the metabolite classes most prominently discussed in today’s literature, future unbiased studies of asthma metabolomics are likely to identify additional important pathways. 

## 2. Short Chain Fatty Acids

Short chain fatty acids (SCFA) are produced by a wide variety of intestinal microbes through fermentation of dietary fiber. The most abundant SCFA are acetate, propionate and butyrate. SCFA exert effects on host physiology by ligation of G-protein coupled receptors including GPR41, GPR43 and GRP109A, and epigenetic modification by inhibition of histone deacetylase [4]. Early evidence was stronger for histone deacetylase inhibitory activity of propionate and butyrate, but a recent study showed that acetate can also inhibit histone deacetylase [5]. SCFA have important immune-modulating properties including induction of T regulatory cell differentiation in mice [6,7,8,9], reduction of eosinophil trafficking and survival [10] and promotion of mucosal antibody production [11].

Accordingly, SCFA are protective against allergic diseases in mouse models including models of pulmonary allergic inflammation and food allergy [12,13,14]. Multiple observational studies in humans have found that reductions in fecal SCFA during infancy are associated with asthma and allergy later in life. In two cohorts of infants, one Canadian and the other Ecuadorian, fecal acetate at age 3 months of age was lower in subjects who later developed atopy and wheeze [15,16]. In another study, European infants in the highest percentile groups of fecal butyrate and propionate abundance had reduced risk of subsequent atopy and asthma [17]. Some murine experimental data and human observational data even suggest that fecal acetate during pregnancy can influence risk of asthma and atopy in offspring [18,19,20]. Microbial metabolism of SCFA may be relevant locally in the airway as well: one study of the bronchial microbiome found an increased predicted capacity for SCFA metabolism in association with asthma [21]. Together, these findings highlight the role of SCFA in the development of asthma and atopy, and suggest that SCFA-directed treatment could be an effective preventive strategy.

## 3. Polyunsaturated Fatty Acids

The major polyunsaturated fatty acid (PUFA) families are omega-3 fatty acids, including α-linolenic acid and its metabolites: eicosapentanoic acid (EPA) and docosahexaenoic acid (DHA); and omega-6 fatty acids, including linoleic acid and its metabolite arachidonic acid. Because omega-6 fatty acids give rise to inflammatory eicosanoids [22] and omega-3 fatty acids displace omega-6 fatty acids in cell membranes and give rise to anti-inflammatory pro-resolving mediators [23,24], a high omega-6 to omega-3 fatty acid ratio is thought to be pro-allergic. Though high-quality evidence is limited with regard to postnatal omega-3 fatty acid supplementation to prevent asthma or allergies [25,26], a promising 30.7% reduction in wheeze at age 3 years was recently reported in offspring of mothers randomized in a clinical trial to receive omega-3 fatty acids during pregnancy [27].

While dietary intake is the dominant source of essential omega-3 and omega-6 fatty acids and PUFA are not synthesized by members of the human microbiota, accumulating evidence points to the importance of PUFA interactions with microbes in asthma pathogenesis. In multiple human studies, fecal PUFA in early life have been inversely associated with asthma and allergy. At age one month, fecal omega-3 docosapentaenoic acid was reduced in infants at risk of atopy or asthma [28] and in a cross-sectional analysis of 3 year-old children, several highly-correlated fecal omega-3 and omega-6 fatty acids were inversely associated with asthma or recurrent wheeze [29]. In a trial of probiotic supplementation with Lactobacillus rhamnosus GG in infants at high risk of asthma, fecal levels of omega-3 fatty acids including docosapentaenoic acid and docosahexaenoic acid were higher in healthy controls and those who had been supplemented with L. rhamnosus GG in comparison to those at risk of asthma who had not received supplementation [30]. In this study, probiotic supplementation appeared to have tolerogenic effects: fecal sterile water samples from 6 month-old infants who had received L. rhamnosus GG induced increased T regulatory cell differentiation and IL-10 production compared to samples from infants who had received a placebo [30].

Several human studies have found that dietary omega-3 fatty acid intake alters intestinal microbiota composition [31,32,33,34,35], and taxa increased in association with omega-3 fatty acid intake have been observed to include producers of SCFA [31,32,33,36]. In addition to potentially increasing SCFA production, PUFA can be metabolized by human gut microbes to produce metabolites including conjugated linoleic acids (CLA) [37,38]. Interestingly, SCFA producers that increase with omega-3 fatty acid intake, such as Bifidobacterium, Lactobacillus and Roseburia spp., are among the most active at metabolizing PUFA to CLA, suggesting a prebiotic effect of omega-3 fatty acids that could include selection for SCFA producers [37,39,40,41,42]. Indeed, CLA consumption or microbial production has also been linked to increased intestinal SCFA [43,44], demonstrating a biochemical link between short and long chain fatty acids.

A few small (n = 28–40 subjects) randomized controlled human trials of CLA supplementation for asthma or allergy provide promising evidence that CLA itself may improve control of existing disease. CLA supplementation in overweight mild asthmatics resulted in weight loss and improved airway hyperresponsiveness [45]. In adults with birch pollen allergy, CLA supplementation reduced sneezing, production of TNF-α, interferon (IFN)-γ and interleukin-5, and release of eosinophil-derived neurotoxin [46]. In children age 6 to 18 years with mild asthma and allergic sensitization, CLA supplementation did not improve pulmonary function or symptoms, but was associated with lower plasma eosinophil cationic protein and peripheral blood mononuclear cell production of IFN-γ and interleukin-4 [47]. Potential mechanisms by which CLA may reduce airway inflammation include peroxisome proliferator-activated receptor-γ (PPARγ) activation [48], GPR40 activation [49] and/or reduction of inflammatory eicosanoid production [50,51].

12,13-dihydroxy-9-octadecenoic acid (12,13-diHOME) is another potentially important metabolite of omega-6 linoleic acid that has been associated with asthma. Peritoneal injection of 12,13-diHOME in mouse models of allergic airways disease led to increased circulating and pulmonary 12,13-diHOME with accompanying airway inflammation, increased IgE and decreased lung T regulatory cells [38]. Effects of 12,13-diHOME appear to be at least partially due to ligation of PPARγ in dendritic cells [38]. In human studies, 12,13-diHOME was elevated in the airways of birch-allergic adult asthmatics after birch challenge [52], and in feces of infants at high risk of subsequent asthma and atopy [28]. This metabolite was linked to microbiota composition via shotgun metagenomics analysis of infant stool samples [38]. Bacterial, but not human, genes encoding epoxide hydrolase enzymes, which catalyze production of 12,13-diHOME, were present at higher abundances in samples from infants at high risk of asthma and atopy. Specific bacterial species were identified that harbored the relevant enzymes and therefore have the capacity to produce 12,13-diHOME. Feeding E. coli engineered to overexpress these epoxide hydrolases led to reduced lung T regulatory cells in a murine allergic airway disease model [38]. 

To summarize, PUFA impact fecal microbiome composition and have been associated with increased production of several metabolites and metabolite classes that impact asthma disease risk, including SCFA, CLA and 12,13-diHOME (Figure 1). However, PUFA have many effects that do not rely on the microbiome, and other factors including dietary intake and genetic variation in PUFA pathway genes such as FADS1/2 have major impacts on PUFA bioavailability [53]. Future studies will need to determine the extent to which microbial metabolic pathways mediate associations between PUFA and asthma development and pathophysiology while also accounting for the roles of dietary and genetic factors.

## 4. Bile Acids

Primary bile acids cholic acid and chenodeoxycholic acid are synthesized in the liver, where they may be conjugated by taurine or glycine and are then secreted into the duodenum. Most bile acids are absorbed distally in the gut and returned to the liver via enterohepatic circulation. Gut microbial enzymatic activity results in production of secondary bile acids such as deoxycholic acid and lithocholic acid [54], and bile acids themselves have antimicrobial activity and can influence microbial composition in the gut [55,56]. 

In vitro and mouse studies show protective effects of bile acids on allergic airway inflammation via multiple mechanisms, including some bile acids that are produced via microbial modification. Ursodeoxycholic acid, a microbially modified bile acid, prevents eosinophilic inflammation in primary biliary cirrhosis [57] and reduces eosinophilic airway inflammation in OVA-sensitized mice via ligation of dendritic cell nuclear farnesoid X receptors [58]. Chenodeoxycholic acid, a primary bile acid, similarly reduces murine allergic airway disease via farnesoid X receptor agonist activity in the lung [59]. Conjugated bile acids, which have not undergone microbial modification, significantly decrease allergen-induced airway inflammatory responses, mucus metaplasia and airway hyperresponsiveness [60]. These effects have been attributed to inhibition of the inflammatory unfolded protein response [60]. Additionally, it was recently found that depletion of gut microbial bile acids leads to reduced gut RORγ+ regulatory T cells via a mechanism involving bile acid activity at the vitamin D receptor [61], though it is not clear if this has an effect on distant organs such as the lung.

Limited evidence from human studies has linked bile acids to asthma. In a birth cohort study, urinary sulfated bile acids glycolithocholate, glycocholenate, and glycohyocholate were elevated and tauroursodeoxycholate was decreased at age 3 months in children who had atopy and wheeze at age 1 year [15]. In a comparison of fecal metabolites in 35 children with asthma and 20 non-atopic controls, significant differences in abundances of taurochenodeoxycholate, taurocholate and glycocholate were found, and there were additional differences in fecal bile acid abundances between subjects with asthma and those with food allergy [62]. Finally, plasma bile acids (taurocholate and glycodeoxycholate) were higher in asthmatic adults than healthy controls and in particular in those with high fractional exhaled nitric oxide, a marker of Th2-high asthma [63]. Nitric oxide itself increases hepatic production and microbial metabolism of bile acids, suggesting that bile acids may serve as biomarkers of the Th2-high asthma endotype [63]. Future studies will be valuable in clarifying the most relevant mechanisms by which bile acids and their modification by microbes impact asthma and whether the bile acid pathway is pertinent to prevention of asthma, morbidity in the setting of existing asthma, or both.

## 5. Tryptophan

Tryptophan is an essential amino acid and can be metabolized via four major pathways to either produce kynurenine derivatives (the major pathway), serotonin derivatives, to be utilized in protein synthesis, or to be metabolized by fecal microbes [64]. Tryptophan metabolism is complex and pathways by which tryptophan is utilized vary by body site and context [64]. Gut microbes are major participants in tryptophan metabolism; an estimated 90% of the serotonin in the body is produced by intestinal microbes [65].

There is evidence that tryptophan metabolites play a role in the pathophysiology of asthma (Figure 2). Indoleamine 2,3-dioxygenase-1 (IDO) metabolizes tryptophan to produce kynurenine derivatives in antigen-presenting cells and other cells resident in lymph nodes and inflammatory tissue. Expression of IDO is induced by IFN-γ and inhibited by Th2 cytokines including IL-4 and IL-13 [66,67]. IDO activity and kynurenine metabolites have anti-inflammatory and tolerogenic properties including reducing T cell inflammation by reducing tryptophan availability [68] and promoting T regulatory cells [69,70]. Interestingly, IDO inhibits growth of intracellular pathogens. So, IDO can both be induced by bacterial motifs via TLR ligation and IFN-γ induction, and in turn can inhibit microbial growth [71,72].

Tryptophan metabolites also impact immune homeostasis via interactions with the aryl hydrocarbon receptor, a ligand-activated transcription factor that senses exposures including polyaromatic hydrocarbons and environmental toxins and impacts transcription of a broad range of genes [73]. The aryl hydrocarbon receptor is expressed in immune cells, gut epithelial cells, and others [73]. Tryptophan metabolites including indole-3-acetate, indole-3-aldehyde, indole, and tryptamine are known to activate the aryl hydrocarbon receptor [74] and many of these metabolites are produced by microbes resident in the human gut [64,75]. Aryl hydrocarbon receptor activation promotes tolerogenic dendritic cells [76], Th17 and T regulatory cell differentiation [77] and impacts innate lymphoid cell (ILC) homeostasis in the gut by stimulating ILC3 cells to produce IL-22 and suppressing ILC2 function including expression of IL-33 receptor, IL-5, IL-13 and amphiregulin [78]. It also boosts gut epithelial barrier function, including response to IL-10 [79,80], though there is some conflicting evidence on this point [81]. Like IDO activity, aryl hydrocarbon receptor activation influences, and is influenced by, microbial composition [64].

Multiple lines of evidence support a protective effect of tryptophan metabolism via the IDO pathway and aryl hydrocarbon receptor activation in asthma. In murine models of asthma, IDO expression induced by activation of TLR9 by bacterial DNA motifs reduce airway hyperreactivity [82] and activation of aryl hydrocarbon receptors reduce airway inflammation and hyperresponsiveness [83,84]. Human studies also support a tolerogenic role for IDO and a reduction in IDO activity in people with asthma. In a study of 205 children, tryptophan and kynurenine levels were higher and IgE and IDO activity lower in those with asthma and allergic rhinitis [85]. In another pediatric population, IDO activity in peripheral blood and induced sputum was lower in children with allergic asthma than healthy controls [86]. This result was more pronounced in children with high FeNO levels. In a study in which subjects with and without asthma were experimentally infected with rhinovirus, though IDO activity was not induced by infection, baseline pulmonary IDO activity was lower and circulating tryptophan and quinolinic acid, a metabolite of the kynurenine pathway, were elevated in asthmatic subjects [87].

Linking these findings to the gut microbiome, in a screen of products produced by probiotics, D-tryptophan was identified as a metabolite produced by Lactobacillus rhamnosus GG and Lactobacillus casei W56 that, when fed to mice, increased lung and gut T regulatory cells and reduced allergic airway disease [88]. In this study, allergic airway disease was associated with reduced gut microbial diversity, and diversity was increased by administration of D-tryptophan. Unlike its enantiomer L-tryptophan, D-tryptophan is a nonproteinogenic metabolite and is produced by numerous bacteria. In addition to having activity at host cell receptors including GPR109B, D-tryptophan can be metabolized by IDO to produce kynurenine metabolites, which may account for its tolerogenic effects [88]. Further supporting a role for gut microbial tryptophan metabolism in allergic disease, human metabolomics studies have linked reductions in fecal tryptophan metabolites to food allergy [62,89]. Additional research is needed to determine the impact of gut microbial tryptophan metabolism on both asthma development and morbidity.

## 6. Sphingolipids

Sphingolipids are bioactive eukaryotic lipids with roles in cell growth regulation, cell–cell interactions, and other cellular functions [90]. Some sphingolipids, especially sphingosine-1-phosphate, have well defined roles in immune function. Specifically, sphingosine-1-phosphate concentration gradients control T cell egress from lymph nodes into circulation [91]. Sphingosine-1-phosphate promotes allergic airway inflammation in mouse models and is elevated in the airways of asthmatic humans after allergen challenge [92,93,94]. 

While sphingosine-1-phosphate appears to promote asthma, other sphingolipid metabolites may be protective. The enzyme encoded by the ORMDL3 gene in the 17q21 region, which is the most replicated childhood asthma genetic locus, inhibits the first step in de novo sphingolipid synthesis [95,96]. A mouse model that overexpresses ORMDL3 exhibits increased airway remodeling and responsiveness and IgE levels [97]. Either administration of myriocin, which, like ORMDL3, inhibits the serine palmitoyltransferase enzyme that initiates sphingolipid synthesis, or heterozygous knockout of the serine palmitoyltransferase gene in mice results in decreased de novo sphingolipid synthesis and increased airway reactivity [98].

A few human studies corroborate preclinical evidence of a link between sphingolipids and asthma. In a sample of pediatric asthmatics, sphingolipids were reduced in those with high-risk variants in the 17q21 locus that promotes expression of ORMDL3 and in those with non-allergic asthma in comparison to those with allergic asthma or healthy controls [99]. De novo sphingolipid synthesis was also lower in children with asthma than controls [99]. In another human study, circulating sphingolipids were inversely associated with childhood asthma and recurrent wheeze, and those with high-risk ORMDL3 expression-promoting genetic variants exhibited limited benefit from vitamin D supplementation in comparison to those with low-risk variants [100]. These results suggest that vitamin D may influence sphingolipid metabolism with protective effects on childhood asthma. In another study of allergen challenge in humans allergic to house dust mite, lung function and airway hyperreactivity correlated with sphingosine-1-phosphate plasma concentrations, which increased after allergen challenge in subjects who developed both early and late phase symptoms [101]. Meanwhile, sphinganine, which is produced in early steps of de novo sphingolipid synthesis, was only increased after allergen challenge in subjects who did not develop an asthmatic response. Together, these findings support the concept that sphingosine-1-phosphate promotes asthma while sphingolipids early in the de novo synthesis pathway may be protective.

Sphingolipids may be dietary, host-derived, or produced by a limited number of microbial taxa, particularly those of the phylum Bacteroidetes. Of interest, bacteria that produce sphingolipids are among the dominant residents of the human gut [102]. Sphingolipids produced by Bacteroides fragilis, in particular, may have relevance to human health as they are ligands of the invariant natural killer cell receptor and modulate invariant natural killer cell recruitment and proliferation in the colon [103]. Accordingly, low fecal sphingolipids in early life have been linked to food allergies in two human studies [62,89]. However, this effect of Bacteroides-derived sphingolipids on invariant natural killer cell homeostasis appears to be limited to the colon, with no effect on asthma susceptibility [103]. Further research is needed to ascertain whether microbial-derived sphingolipids may affect asthma pathophysiology via other mechanisms.

## 7. Conclusions

Microbiome–metabolite interactions are pervasive in the human body and have relevance to many human diseases, including asthma. Among microbial-derived metabolites, the evidence is strongest that SCFA, PUFA and bile acids contribute to asthma pathophysiology. Sphingolipids and tryptophan metabolites are worthy of future research as potentially important pathways. Of these classes, some including SCFA and 12,13-diHOME appear to play a role early in life before the onset of disease, while others including CLA and tryptophan metabolites of the kynurenine pathway are best studied in the context of existing asthma. These findings provide a rationale for the development of microbe- and metabolite-targeted treatments for asthma and other diseases. Strategies include probiotics, prebiotics and other dietary modifications, supplementing or inhibiting microbial-derived metabolites, and fecal microbiome transplant [104]. As understanding of metabolite–microbe interactions continues to grow, we expect additional insights to guide precision medicine approaches to health and disease.

## Figures and Tables

**Figure 1 metabolites-10-00097-f001:**
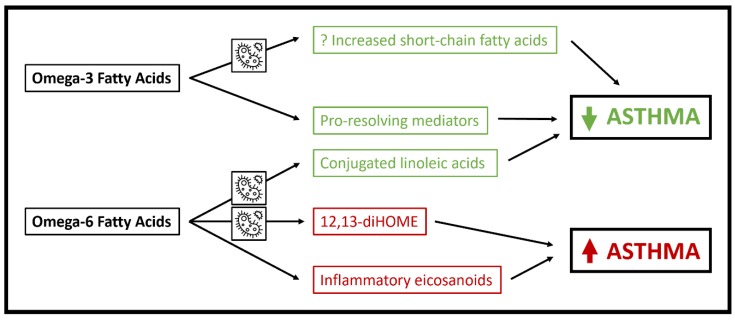
Schematic of mechanisms whereby polyunsaturated fatty acids may influence asthma pathophysiology. Pathways in which microbial metabolism is essential are highlighted by the microbe icon.

**Figure 2 metabolites-10-00097-f002:**
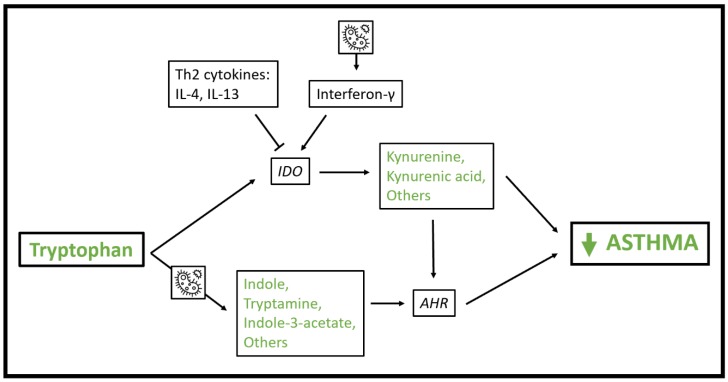
Schematic of mechanisms whereby tryptophan metabolic pathways influence asthma pathophysiology. As indicated by the microbe icon, microbial exposure induces interferon-γ production, and gut bacteria participate in metabolism of tryptophan to indole, tryptamine and other metabolites with activity at the aryl hydrocarbon receptor. Abbreviations: AHR = aryl hydrocarbon receptor; IDO = Indoleamine 2,3-dioxygenase-1.

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
