# Peer review of "Gut Microbial-Derived Metabolomics of Asthma"

_metabolites, 2020, doi:10.3390/metabo10030097_

Round 1

Reviewer 1 Report

This is an excellent review discussing gut-microbiome derived alterations in asthmatic metabolome. 

Although, it's a tough task, authors should provide one or two illustrations integrating the major metabolites with direct impact and therapeutic applications. This is a minor comment.

Author Response

Point 1: This is an excellent review discussing gut-microbiome derived alterations in asthmatic metabolome. 

Although, it's a tough task, authors should provide one or two illustrations integrating the major metabolites with direct impact and therapeutic applications. This is a minor comment.

Response 1:

Thank you for your kind review! We agree that a visual illustration of the more complex pathways would be helpful to readers and added schematics showing how PUFA metabolites and tryptophan metabolites impact asthma pathophysiology with the figures starting at lines 136 and 229, respectively.

Reviewer 2 Report

In this review article the authors discuss and review the gut derived metabolites associated with asthma and the role of immune modulation. This is a topical area of research and the authors provide some insights which could be important since there is continuing development of approaches for harnessing microbial metabolites for therapeutic applications. Some issues and concerns that need to be addressed are indicated below.

1.       It should be noted that although acetate is  included as a SCFA there has not been strong evidence for its activity as an HDAC inhibitor. However, recent studies clearly show that acetate is an HDAC inhibitor (Sci Rep. 7, 10163, 2017).

2.       Define 12,13-diHOME prior to using the acronym.

3.       The tryptophan metabolite section could be expanded since there numerous studies on the health protecting role of AhR-active compounds. Moreover, there is also a mechanistic underpinning to the role of AhR ligands and their protective effects in the gut.

Author Response

In this review article the authors discuss and review the gut derived metabolites associated with asthma and the role of immune modulation. This is a topical area of research and the authors provide some insights which could be important since there is continuing development of approaches for harnessing microbial metabolites for therapeutic applications. Some issues and concerns that need to be addressed are indicated below.

Point 1: It should be noted that although acetate is  included as a SCFA there has not been strong evidence for its activity as an HDAC inhibitor. However, recent studies clearly show that acetate is an HDAC inhibitor (Sci Rep. 7, 10163, 2017).

Response 1: 

Thank you for this comment. We added the following text at lines 47-79:

"Early evidence was stronger for histone deacetylase inhibitory activity of propionate and butyrate, but a recent study showed that acetate can also inhibit histone deacetylase [5]"

Ref 5: Jin U.H.; Cheng, Y.; Park, H.; Davidson, L.A.; Callaway, E.S.; Chapkin, R.S.; Jayaraman, A.; Asante, A.; Allred, C.; Weaver, E.A.; et al. Short chain fatty acids enhance aryl hydrocarbon (Ah) responsiveness in mouse colonocytes and Caco-2 human colon cancer cells. Sci Rep 2017, 7, 10163.

Point 2: Define 12,13-diHOME prior to using the acronym.

Response 2: 

We replaced the first mention of 12,13-diHOME at line 114 with "12,13-dihydroxy-9-octadecenoic acid (12,13-diHOME)".

Point 3: The tryptophan metabolite section could be expanded since there numerous studies on the health protecting role of AhR-active compounds. Moreover, there is also a mechanistic underpinning to the role of AhR ligands and their protective effects in the gut.

Response 3: 

We agree that this section warrants further elaboration and added the bolded text below to lines 190-202:

"Tryptophan metabolites also impact immune homeostasis via interactions with the aryl hydrocarbon receptor, a ligand-activated transcription factor that senses exposures including polyaromatic hydrocarbons and environmental toxins and impacts transcription of a broad range of genes[73]. The aryl hydrocarbon receptor is expressed in immune cells, gut epithelial cells, and others[73]. Tryptophan metabolites including indole-3-acetate, indole-3-aldehyde, indole, and tryptamine are known to activate the aryl hydrocarbon receptor[74] and many of these metabolites are produced by microbes resident in the human gut[64,75]. Aryl hydrocarbon receptor activation promotes tolerogenic dendritic cells[76], Th17 and T regulatory cell differentiation[77] and impacts innate lymphoid cell (ILC) homeostasis in the gut by stimulating ILC3 cells to produce IL-22 and suppressing ILC2 function including expression of IL-33 receptor, IL-5, IL-13 and amphiregulin[78]. It also boosts gut epithelial barrier function, including response to IL-10[79,80], though there is some conflicting evidence on this point [81]. Like IDO activity, aryl hydrocarbon receptor activation influences and is influenced by microbial composition[64]."

New references:

  1. veit Rothhammer; Quintana, F.J. The aryl hydrocarbon receptor: an environmental sensor integrating immune responses in health and disease. Nat Rev Immunol.201919, 184-97.
  2. Ettmayer, P.; Mayer, P.; Kalthoff, F.; Neruda, W.; Harrer, N.; Hartmann, G.; Epstein, M.M.; Brinkmann, V.; Heusser, C.; Woisetschläger M. A novel low molecular weight inhibitor of dendritic cells and B cells blocks allergic inflammation. Am J Respir Crit Care Med 2006,173, 599-606.
  3. Metidji, A.; Omenetti, S.; Crotta, S.; Li, Y.; Nye, E.; Ross, E.; Li, V.; Maradana, M.R.; Schiering, C.; Stockinger, B. The environmental sensor AHR protects from inflammatory damage by maintaining intestinal stem cell homeostasis and barrier integrity. Immunity.200950, 1542.
  4. Lanis, J.M.; Alexeev, E.E.; Curtis, V.F.; Kitzenberg, D.A.; Kao, D.J.; Battista, K.D.; Gerich, M.E.; Glover, L.E.; Kominsky, D.J.; Colgan, S.P. Tryptophan metabolite activation of the aryl hydrocarbon receptor regulations IL-10 receptor expression on intestinal epithelia. Mucosal Immunol.201710, 1133-44.
  5. Iyer, S.S.; Gensollen, T.; Gandhi, A.; Oh, S.F.; Neves, J.F.; Collin, F.; Lavin, R.; Serra, C.; Glickman, J.; de Silva, P.S.A.; et al. Dietary and microbial oxazoles induce intestinal inflammation by modulating aryl hydrocarbon receptor responses. Cell2018173, 1123-34.e11.